# Butyrate Protects Myenteric Neurons Loss in Mice Following Experimental Ulcerative Colitis

**DOI:** 10.3390/cells12131672

**Published:** 2023-06-21

**Authors:** Marcos A. F. Caetano, Henrique I. R. Magalhães, Jheniffer R. L. Duarte, Laura B. Conceição, Patricia Castelucci

**Affiliations:** 1Department of Anatomy, Institute of Biomedical Sciences, University of São Paulo, São Paulo 05508-000, Brazil; marcosaferf@usp.br (M.A.F.C.); jhenifferlima.duarte@usp.br (J.R.L.D.); laurabarbosa@usp.br (L.B.C.); 2Department of Surgery, School of Veterinary Medicine and Animal Sciences, University of São Paulo, São Paulo 05508-270, Brazil; inhauser@usp.br

**Keywords:** inflammatory bowel diseases, short-chain fatty acids, gut microbiota, Butyrate, enteric nervous system, free fatty acid receptor 3, GPR41

## Abstract

The enteric nervous system is affected by inflammatory bowel diseases (IBD). Gut microbiota ferments dietary fibers and produces short-chain fatty acids, such as Butyrate, which bind to G protein–coupled receptors, such as GPR41, and contribute to maintaining intestinal health. This work aimed to study the GPR41 in myenteric neurons and analyze the effect of Butyrate in mice submitted to experimental ulcerative colitis. The 2, 4, 6 trinitrobenzene sulfonic acid (TNBS) was injected intrarectally in C57BL/6 mice (Colitis). Sham group received ethanol (vehicle). One group was treated with 100 mg/kg of Sodium Butyrate (Butyrate), and the other groups received saline. Animals were euthanized 7 days after colitis induction. Analyzes demonstrated colocalization of GPR41 with neurons immunoreactive (-ir) to nNOS and ChAT-ir and absence of colocalization of the GPR41 with GFAP-ir glia. Quantitative results demonstrated losses of nNOS-ir, ChAT-ir, and GPR41-ir neurons in the Colitis group and Butyrate treatment attenuated neuronal loss. The number of GFAP-ir glia increased in the Colitis group, whereas Butyrate reduced the number of these cells. In addition, morphological alterations observed in the Colitis group were attenuated in the Butyrate group. The presence of GPR41 in myenteric neurons was identified, and the treatment with Butyrate attenuated the damage caused by experimental ulcerative colitis.

## 1. Introduction

The gastrointestinal tract (GIT) has an extensive intrinsic nervous system called the enteric nervous system (ENS), which is made up of interconnected networks of neurons, axons, and enteric glial cells [1]. The ENS has two plexuses: the myenteric (Auerbach’s plexus) and the submucosal (Meissner’s plexus) [2]. The myenteric plexus controls intestinal motility, and the submucosal plexus regulates blood and fluid flow in the mucosa and the secretion of digestive substances and intestinal hormones [2].

Inflammatory Bowel Diseases (IBD) comprise Crohn’s Disease (CD) and Ulcerative Colitis (UC), which are recurrent diseases characterized by an inflammatory condition in the GIT [3]. CD and UC are public health problems that considerably impair the quality of life of patients, entail high costs for health services, and can progress to complications, such as cancer and even death [4]. Although the pathogenesis of IBD remains uncertain, many studies indicate that the dysregulation of the immune system and intestinal microbiota, together with genetic factors, may be related to the development of these diseases [5,6,7].

IBD affects the ENS and, consequently, good intestinal functioning. Neuronal degeneration, necrosis, and apoptosis have been observed in experimental models of ulcerative colitis [8,9,10,11]. In addition to morphological changes, colitis also promotes changes in the expression of neurotransmitters and their receptors, consequently altering the chemical code of neurons [12,13].

Under physiological conditions, the bacteria of the intestinal microbiota are capable of promoting the fermentation of dietary fibers and producing short-chain fatty acids. Acetate, propionate, and butyrate are the most abundant SCFA produced and have a beneficial role in the intestine [14]. Once produced, most SCFA is absorbed by colonocytes, mainly through active transport by monocarboxylate transporters 1 (MCT1), sodium-dependent monocarboxylate transporters (SMCT1), and passive diffusion [15,16,17]. After being absorbed, SCFA is converted into ATP, which is used as an energy source by these cells [18]. The remaining unmetabolized SCFA passes through the basolateral membrane and reaches the portal circulation, being an energy source for hepatocytes and in the cholesterol synthesis [19,20]. Only a small amount of SCFA not used by the liver reaches the systemic circulation [21].

SCFA can also bind to G protein–coupled free fatty acid receptors present in the GI tract, CNS, and in various organs and tissues, such as the heart, kidneys, spleen, pancreas, and adipose tissue [16,22]. The GPR41 receptor, also known as FFAR3, is one of the main receptors for SCFAs and, when activated, promotes G-Protein-mediated intracellular signaling cascades [16,23,24]. Among the SCFA capable of activating GPR41 receptors, Butyrate stands out due to its pronounced anti-inflammatory, antioxidant, and neuromodulatory characteristics [25,26].

Is known that the ENS can respond to stimuli through SCFA produced by bacteria and that there is the presence of the GPR41 receptor [27]. This suggests that this SCFA may have therapeutic potential in the treatment of intestinal diseases, such as ulcerative colitis [28]. However, it is not known in which types of enteric neurons the GPR41 receptor is found [29] and what are the effects of this receptor and SCFA on enteric neurons in the face of intestinal inflammation [30]. In this context, the objective of this work was to study the GPR41 receptor in neurons of the myenteric plexus, specifically in immunoreactive neurons (-ir) to the enzyme neuronal nitric oxide synthase (nNOS), choline acetyltransferase (ChAT) and enteric glial cells glial fibrillary acid protein (GFAP)-ir, and analyze the effects of the use of Butyrate in mice submitted to experimental ulcerative colitis, with the purpose of evaluating whether Butyrate has a beneficial effect on ulcerative colitis and enteric neurons.

## 2. Materials and Methods

### 2.1. Animals

This work was conducted according to current regulations of the Ethics Committee on Animal Use of the Biomedical Science Institute of the University of São Paulo. All procedures were approved by the Ethics Committee on Animal Use of the Biomedical Science Institute of the University of São Paulo (Protocol 6507140420). Fifteen male wild-type C57BL/6 mice weighing 20–26 g, 8 weeks old, were used. These animals were maintained under standard conditions at 21 °C and a 12 h light–dark cycle, with food and water ad libitum.

### 2.2. Experimental Ulcerative Colitis

The animals were randomly allocated into three groups: Sham, Colitis, and Butyrate. Each group had *n* = 5 animals. For induction of experimental ulcerative colitis, 8-week-old male wild-type C57BL/6 mice were anesthetized with xylazine (10 mg/kg of animal weight) and ketamine (100 mg/kg of animal weight) by subcutaneous injection. After being anesthetized, the animals in the Colitis and Butyrate groups received an intrarectal injection of 100 μL of 2, 4, 6, trinitrobenzene sulfonic acid (TNBS) (Sigma-Aldrich, St. Louis, MO, USA) 1.5% diluted in ethanol 35%, with the aid of a 4 cm polypropylene cannula. Animals in the Sham group received only 35% ethanol (TNBS vehicle) intrarectally [8,9,10]. The animals in the Butyrate groups received a daily solution of Sodium Butyrate (Sigma, Saint Louis, MO, USA) (100 mg/kg per day), dissolved in 0.9% NaCl saline solution, via oral gavage, for 7 days. The first Sodium Butyrate gavage was performed on the same day as the TNBS injection after the animals had recovered from anesthesia. Sham and Colitis animals received an equivalent volume of saline during the same period [31]. After 7 days of TNBS or alcohol injection, the animals were anesthetized with anesthetic overdose, and the colon was collected and had its length measured. At the time of collection, a macroscopic analysis was performed using a scoring scheme [32]. The scores were stratified as 0—Normal; 1—Presence of hyperemia without ulcers; 2—Ulcerations without hyperemia; 3—Ulcerations at one site; 4—Two or more sites of ulcerations; 5—Sites of damage extending >1 cm; and 6–10—Sites of damage extending >2 cm, with the score increasing by 1 for each additional cm.

Throughout the experimental period, the animals were weighed daily, always at the same time, and the stool contained in the cages was analyzed daily for consistency and the macroscopic presence of blood in order to evaluate the Disease Activity Index (DAI). The DAI was obtained by adding the scores, for each criterion, of the animals in each group, according to Table 1 [33,34,35].

### 2.3. Immunofluorescence

Distal colon tissues were opened at the mesenteric border, cleaned with 0.01 M phosphate-buffered saline (PBS), and placed mucosal side down on a sheet of balsa wood. Subsequently, the tissues were immersed for fixation in 4% paraformaldehyde in 0.1 M sodium phosphate buffer (pH 7.3) at 4 °C for 24 h. This procedure was performed by the same researcher to maintain a homogeneous tension in all preparations. The next day, the tissues were removed from the fixative and cleared with 3 washes of 10 min in Dimethyl sulfoxide P.A., followed by 3 washes of 10 min in 0.01 M PBS. Then, the tissues were stored in PBS sodium-azide 0.1% at 4 °C. Subsequently, the tissues of each animal were dissected, removing the mucous layer together with the submucosal plexus and the circular muscle layer to obtain the whole mounts, which consists of the longitudinal muscle associated with the myenteric plexus [8,10].

For immunofluorescence, the preparations were immersed in 10% normal horse serum solution in PBS containing 1.5% Triton X-100 (Sigma, USA) for 45 min at room temperature. After, the tissues were incubated with the primary antibodies described in Table 2 for 48 h at 4 °C. The choice of these markers was based on works that characterize the chemical code of enteric neurons and glia [1]. nNOS is a marker of inhibitory motor neurons, ChAT is a marker of excitatory cholinergic neurons, PGP9.5 is a pan-neuronal marker, and GFAP is an enteric glial cell marker.

After incubation with the primary antibody, the whole mounts were washed 3 times for 10 min each in 0.01 M PBS and was incubated in a combination of secondary antibodies described in Table 2 for 1 h at room temperature. Then, the tissues were washed 3 times for 10 min each in 0.01 M PBS. After, they were incubated with 4′, 6-diamidino-2-phenylindole (DAPI) for 3 min, washed 3 times for 5 min in 0.01 M PBS, and mounted on slides with glycerol buffered with 0.5 M calcium carbonate (pH 8.6) [10]. The preparations were examined with a fluorescence microscope Nikon 80i (Tokyo, Japan), and images were captured using a digital camera and the software NIS-elements AR 3.1 (Nikon). Additionally, the preparations were analyzed and photographed with LSM 780-NLO Laser Scanning Confocal Microscope (Zeiss). The images of preparations obtained from the confocal microscope were processed, and the boards were made using CorelDraw Graphic Suite 2020 software.

### 2.4. Qualitative Analysis

For colocalization studies, enteric neurons and glial cells were identified by immunofluorescence. The labeling of the second antigen was evaluated using a second filter, and subsequently, the two antigens were superimposed to assess colocalization.

### 2.5. Quantitative Analysis

For the number of neurons and glial cells per ganglion, 50 ganglia were counted for each labeling of neurons GPR41-ir, nNOS-ir, ChAT-ir, and GFAP-ir glial cells of each animal (*n* = 5 animals/group) with a Nikon 80i fluorescence microscope at a 40× objective. The results were expressed in the number of neurons per ganglion.

To obtain the neuronal profile area, 125 nNOS-ir neurons and 125 ChAT-ir neurons and their respective nuclei of each animal (*n* = 5 animals/group) were photographed. The images were captured by the NIS-elements AR 3.1 program (Nikon), and the cell profile area analyses were performed using the Image Pro Plus 5.0 program. The area of the cell bodies of the nNOS-ir and ChAT-ir neurons and their respective DAPI-labeled nuclei was measured. The cytoplasmic area was calculated by subtracting the neuronal body area from the nucleus area. The results were expressed in µm^2^.

Corrected Total Cell Fluorescence (CTCF) of neurons GPR41-ir, nNOS-ir, ChAT-ir, and GFAP-ir enteric glia was obtained with Image J^®^ software version 1.46r (National Institutes of Health, Bethesda, MD, USA). Photomicrographs captured by the NIS-elements AR 3.1 program (Nikon) were used to obtain the parameters “area”, “integrated density”, and “mean gray value” in 20 ganglia (*n* = 5 animals/group), and the correction factor was obtained through the average of 5 background measurements in each ganglion. The values obtained were used to calculate the CTCF through the following (Equation (1)) [36]:CTCF = Integrated Density − (Area × Mean Background Fluorescence)(1)

### 2.6. Histological Analysis

Distal colon tissues measuring 1.5 cm in length were collected from 5 animals in each group. The tissues were opened at the mesenteric border, cleaned with 0.01 M PBS, and placed mucosal side down on a sheet of balsa wood. Subsequently, the tissues were immersed for fixation in 4% paraformaldehyde in 0.1 M sodium phosphate buffer (pH 7.3) at 4 °C for 48 h. The tissues were dehydrated in batteries with increasing concentrations of alcohol, cleared in 3 xylene batteries, and then embedded in paraffin blocks. Subsequently, transverse sections of the tissues, with a thickness of 5 μm, were performed using a microtome. Tissues were stained with Hematoxylin & Eosin (H&E) and Periodic Acid Schiff (PAS). Qualitative analyzes were also carried out with a Nikon 80i microscope coupled to a camera with NIS-elements AR 3.1 software (Nikon).

A microscopic evaluation of colitis was performed using a scoring system [37,38]. The scores were categorized as follows according to each corresponding parameter: Ulcerations: 0—No ulcer; 1—Single ulceration not exceeding the muscularis mucosa; 2—Ulcerations not exceeding the mucosa; 3—Ulcerations exceeding the submucosa. Submucosal edema: 0—No edema; 1—Mild edema; 2—Moderate edema; 3—Severe edema. Inflammatory cell infiltration: 0—No infiltration; 1—Mild infiltration; 2—Moderate infiltration; 3—Dense infiltration.

To obtain the number of goblet cells, counts were made in 10 intestinal crypts in 4 semi-sequential histological sections (10 µm of the interval between sections) per animal (*n* = 5 animals per group) through PAS staining [39].

### 2.7. Statistical Analysis

The results obtained were expressed as mean ± standard error. Data from all groups were analyzed, and the comparison between them was performed using the one-way ANOVA analysis of variance, followed by the Tukey Test, using the GraphPad Prism 8.0 software. The tests were performed at the significance level of *p* < 0.05.

## 3. Results

### 3.1. Experimental Ulcerative Colitis

After the TNBS or ethanol injection, the animals were weighed daily, always at the same time, for 7 days. On the first day of the experimental protocol, the animals in the Colitis group showed an average score of 3.0 points, while the animals in the Sham group had an average of 0.6 points, and the animals in the Butyrate group had an average of 2.4 points. On the last day of the experimental protocol, the animals in the Colitis group still had an average of 2.0 points, while the animals in the Sham group showed an average of 0.0 points, and the animals in the Butyrate group had an average of 1.2. In general, the animals in the Colitis group had greater weight variations than those in the Sham group, and the Butyrate group had lower scores when compared to the Colitis group (Figure 1A).

The analysis of stool and/or rectal bleeding showed that only the Colitis group scored bleeding scores on the first and second day after TNBS injection. The animals in the Sham and Butyrate groups had no bleeding in the stool (Figure 1B). Regarding the analysis of feces consistency, the animals in the Colitis group obtained an average score of 2.0 points on the first day after colitis induction and 1.0 points on the second day after induction. On the other days, the consistency of the stool returned to normal. The animals in the Butyrate group had lower scores when compared to the Colitis group, obtaining 1.0 points only on the first day of colitis induction, returning to normal stool consistency from day 2. The animals in the Sham group did not show changes in the consistency of the stools and, therefore, did not score. (Figure 1C). Regarding the length of the large intestine, the mean size of the large intestine in the Colitis group (5.1 ± 0.2 cm) was 14.4% smaller when compared to the Sham group (5.9 ± 0.1 cm; *p* < 0.006) and 14.0% higher in the Butyrate group when compared to the Colitis group (5.9 ± 0.1 cm; *p* < 0.006) (Figure 1D).

Macroscopic analysis of the collected intestines only demonstrated the identification of hyperemia in the Colitis group, which scored 1.0 points. In the Sham group and the Butyrate group, no macroscopic changes were identified.

### 3.2. Qualitative Analysis

In the analysis of the GPR41-ir receptor neurons, it was observed that the label was present in the plasma membrane of the neurons. In the analysis of the triple markings, it was observed that there was colocalization of the GPR41-ir receptor with nNOS-ir neurons (Figure 2), ChAT-ir neurons (Figure 3), and PGP9.5-ir neurons (Figure 4). In the analysis of the triple labeling of the GFAP-ir enteric glia, there was no colocalization with the GPR41 receptor (Figure 5).

### 3.3. Quantitative Analysis

The number of GPR41-ir neurons per ganglion showed a 28.8% reduction in the Colitis group (15.8 ± 0.2) compared with the Sham group (22.2 ± 0.2; *p* < 0.0001) and an increase of 25.5% in the Butyrate group (21.2 ± 0.2; *p* < 0.0001) when compared to the Colitis group (Figure 6A). Analysis of the number of nNOS-ir neurons per ganglion showed a 31.7% reduction in the Colitis group (4.1 ± 0.1) compared to the Sham group (6.0 ± 0.2; *p* < 0.0001). In the Butyrate group, the number of nNOS-ir neurons increased by 31.7% when compared to the Colitis group (6.0 ± 0.1; *p* < 0.0001) (Figure 6B). The number of ChAT-ir neurons per ganglion showed a reduction of 28.3% in the Colitis group (6.1 ± 0.1) when compared to the Sham group (8.6 ± 0.1; *p* < 0.0001) and an increase of 26.2% in the Butyrate group (8.4 ± 0.1; *p* < 0.0001), compared to the Colitis group (Figure 6C). The number of GFAP-ir glia per ganglion showed an increase of 34.5% in the Colitis group (20.6 ± 0.1) when compared to the Sham group (15.3 ± 0.1; *p* < 0.0001) and a reduction of 24.20% in the Butyrate group (15.7 ± 0.2; *p* < 0.0001), compared to the Colitis group (Figure 6D).

The analysis of the neuronal profile area (μm^2^) of the nNOS-ir neurons showed a 9.45% increase in the area of neurons in the Colitis group (234.0 ± 3.8 μm^2^) when compared to the Sham group (213.8 ± 1.5 μm^2^; *p* < 0.001) and a 7.05% reduction in the area of these neurons in the Butyrate group (217.5 ± 0.9 μm^2^; *p* < 0.01) when compared to the Colitis group (Figure 7A). Analysis of the nuclear area of nNOS-ir neurons showed no difference (*p* > 0.05) between the Sham (100.7 ± 1.4 μm^2^), Colitis (103.9 ± 1.5 μm^2^), and Butyrate (102.5 ± 1.9 μm^2^) group (Figure 7B). In order to obtain the cytoplasmic area of these neurons, the total area of the nNOS-ir neurons was subtracted from the area of their respective nuclei. There was an increase of 12.8% in the cytoplasmic area in the Colitis group (130.1 ± 2.7 μm^2^) when compared to the Sham group (113.4 ± 2.1 μm^2^; *p* < 0.001) and a reduction of 11.6% in the cytoplasmic area of the Butyrate group (115 ± 2.4 μm^2^; *p* < 0.01) when compared to the Colitis group (Figure 7C).

Analysis of the neuronal profile area of ChAT-ir neurons showed a 14.8% reduction in the area of neurons in the Colitis group (213.5 ± 1.2 μm^2^) when compared to the Sham group (250.6 ± 4.2 μm^2^; *p* < 0.001) and a 15.1% increase in the area of these neurons in the Butyrate group (251.6 ± 1.8 μm^2^; *p* < 0.001) when compared to the Colitis group (Figure 7D). Regarding the analysis of the nuclear area of ChAT-ir neurons, no difference (*p* > 0.05) was observed between the Sham (104.8 ± 2.4 μm^2^), Colitis (102.0 ± 2.5 μm^2^), and Butyrate (109.1 ± 2.7 μm^2^) (Figure 7E). To obtain the cytoplasmic area of these neurons, the total area of ChAT-ir neurons was subtracted from the area of their respective nuclei. There was a 23.5% reduction in the cytoplasmic area in the Colitis group (111.5 ± 3.5 μm^2^) when compared to the Sham group (145.8 ± 5.6 μm^2^; *p* < 0.001), and an increase of 21.8% in the cytoplasmic area of the Butyrate group (142.5 ± 1.8 μm^2^; *p* < 0.001) when compared to the Colitis group (Figure 7F).

The percentage of distributions of nNOS-ir neuron size ranged from 100 to 500 μm^2^ of total area, from 60 to 170 μm^2^ for the nuclear area of nNOS-ir neurons, and from 50 to 350 μm^2^ for the cytoplasmic area of nNOS-ir neurons (Figure 8). Regarding the percentage of distributions of ChAT-ir neuron size ranged from 150 to 400 μm^2^ of total area, from 60 to 160 μm^2^ for the nuclear area of ChAT-ir neurons, and from 50 to 250 μm^2^ for the cytoplasmic area of ChAT-ir neurons (Figure 9).

The analysis of GPR41-ir myenteric neurons showed a reduction in CTCF of 29.8% in the Colitis group (541,127 ± 35,467) when compared to the Sham group (771,025 ± 80,560; *p* < 0.05) and an increase of 37.0% in the Butyrate group (859,471 ± 39,029) when compared to the Colitis group (*p* < 0.005) (Figure 10A). Regarding nNOS-ir myenteric neurons, there was no statistical difference between groups (*p* > 0.05) (Figure 10B). Myenteric ChAT-ir neurons showed a reduction in CTCF of 21.0% in the Colitis group (1,530,379 ± 52,098) when compared to the Sham group (1,937,090 ± 53,896; *p* < 0.001) and an increase of 21.3% in the Butyrate group (1,944,830 ± 53,965) when compared to the Colitis group (*p* < 0.001) (Figure 10C). GFAP-ir glial analysis demonstrated an increase in CTCF of 47.6% in the Colitis group (2,636,322 ± 136,587) when compared to the Sham group (1,381,989 ± 107,950; *p* < 0.001) and a reduction of 39.8% in the Butyrate group (1,588,239 ± 32,304) when compared to the Colitis group (*p* < 0.001) (Figure 10D).

### 3.4. Histological Analysis

Histological analysis using H&E staining revealed that in the Colitis group, there was a loss of mucosal integrity, the intestinal crypts were misshapen, with foci of epithelial destruction, hyperemia of blood vessels, submucosal edema, and a mild inflammatory infiltrate in the own blade. In addition, cellular changes were observed in myenteric neurons, such as the presence of cytoplasmic vacuoles and changes in nuclear size. In the Sham group, the findings are consistent with typical morphology of the distal colon, with intestinal crypts with preserved architecture, characteristic lamina propria, absence of submucosal edema, inflammatory infiltrate, and hyperemia. Furthermore, the myenteric ganglia presented normal morphology. In the Butyrate group, it was observed that the architecture of the intestinal crypts was more preserved than in the Colitis group, and the destruction of epithelial cells was attenuated. Submucosal edema, inflammatory infiltrate, and hyperemia were not observed in this group. There were also no morphological alterations in the myenteric ganglia, and the Butyrate group was similar to the Sham group (Figure 11).

Microscopic analysis of experimental ulcerative colitis was performed through histology slides stained with H&E. Considering the criteria for microscopic analysis; scores were attributed to each of the groups in each criterion analyzed. None of the groups presented ulcerations and, therefore, did not score any score in this category. As for the analysis of submucosal edema, the Colitis group presented mild submucosal edema, assigning 1 point, and the Sham and Butyrate groups did not present submucosal edema and, therefore, did not score any score in this category. As for the presence of inflammatory infiltrate, the Colitis group presented mild inflammatory infiltrate, assigning 1 point, and the Sham and Butyrate groups did not present an inflammatory infiltrate and, therefore, did not score in this category.

In the PAS staining, it was possible to observe that the goblet cells in the Colitis group were dysmorphic, appeared to have less mucous content, and were in smaller quantities. In the Sham group, goblet cells were quite numerous, with typically normal morphology. In the Butyrate group, the goblet cells presented similar morphology to the Sham group, appearing to be in greater numbers and with a greater apparent amount of mucus than compared to the Colitis group (Figure 12).

The analysis of the number of goblet cells showed a reduction of 31.40% of this cell population in the Colitis group (8.34 ± 0.05) when compared to the Sham group (12.15 ± 0.31; *p* < 0.0001) and an increase of 29.36% in the Butyrate group (11.80 ± 0.46; *p* < 0.0001) when compared to the Colitis group (Figure 13).

## 4. Discussion

Administration of TNBS diluted in 35% ethanol triggers intestinal inflammation mainly by disrupting the intestinal barrier, as ethanol breaks the epithelial layer and exposes the lamina propria to TNBS itself and intestinal lumen antigens, including microorganisms [40]. Although the pathogenesis of IBD remains uncertain, there is evidence that the development of IBD may be related to dysregulation of the immune system and intestinal microbiota in genetically susceptible patients [41,42]. In individuals with IBD, there is a dysregulation of the intestinal barrier, leading to greater intestinal permeability. This can lead to immune responses against microorganisms of the intestinal microbiota, such as the production of anti-*Saccharomyces cerevisiae* antibodies (ASCA) and the production of perinuclear antineutrophil cytoplasmic autoantibodies (pANCA) [43,44]. ASCA and pANCA have been considered physiological markers for IBDs. In particular, ASCA is more associated with CD, and pANCA is more associated with UC [43]. These two markers can be used for the differential diagnosis between CD and UC [45], although they are not exclusive markers of IBD since ASCA is also present in patients with celiac disease [43,44]. In this context, a better understanding of the role of the immune system in response to changes in the intestinal barrier can help to better understand the development of IBD and guide new therapies.

Some clinical studies have been carried out with Sodium Butyrate supplementation for the treatment of IBD, obtaining results of improvement in fecal calprotectin levels, disease activity index, and maintenance of remission of these diseases [46]. The main challenges for the implementation of Butyrate in clinical practice are related to the standardization of an effective dose and the development of a pharmaceutical form that optimizes the administration and adherence of Butyrate as a drug [47]. Given the beneficial effects of Butyrate, demonstrated in several analyzes of this work, the therapeutic potential of this SCFA is reinforced as a possible treatment/adjuvant for IBD, with an additional perspective of improvement in the damage caused to the ENS. These results open perspectives for future studies about the mechanisms of action involved in neuronal loss and the activation of the GPR41 receptor by Butyrate.

The literature reports the presence of GPR41 receptors in the ENS, whose SCFA can bind and trigger effects that can alter patterns of motility, hormone secretion, and communication with the immune system [27,48]. In this work, through qualitative analyses, it was demonstrated the colocalization of the GPR41 Receptor with nNOS-ir and ChAT-ir neurons. However, there was no colocalization of GFAP-ir enteric glia with the GPR41 receptor. Although they supposedly do not have the GPR41 receptor, glia may have other SCFA receptors or even interact with SCFA directly through their entry into these cells through monocarboxylate-type transporters [24,49].

In addition to clinical, macroscopic, and microscopic changes at the mucosal level, several studies have reported that IBD causes changes in the ENS, such as a reduction in the number of enteric neurons, degeneration, necrosis, apoptosis, and changes in intestinal motility [8,9,50]. The quantitative analysis identified a reduction in the number of nNOS-ir, ChAT-ir, and GPR41-ir neurons in the Colitis group when compared to the Sham group. These results corroborate data from the literature that document the reduction of enteric neurons in IBD protocols [8,9,10,51]. In this study, after 7 days of acid injection, neuronal reduction was observed. This can be explained because, even in mild conditions, IBD leads to a loss of intestinal barrier function and changes in neurons and ENS fibers [2]. These alterations may persist even after the resolution of intestinal inflammation, as there is prolonged hyperexcitability of enteric neurons, which disrupts the intestinal motility [50].

The reduction of nNOS neurons was greater than that of ChAT neurons. This finding may be because the inflammatory stimulus promotes an abnormal increase in the intracellular Ca^2+^ [52,53]. Particularly in nNOS-ir neurons, this increase in intracellular Ca^2+^ leads to the production of NO radicals, which, adding to the reactive oxygen species, produce peroxynitrites that culminate in cellular degradation and lipid peroxidation and, therefore, make this neuronal class more susceptible to damage than other neuronal classes [54]. On the other hand, the treatment of animals with Sodium Butyrate was able to attenuate the neuronal loss resulting from intestinal inflammation, maintaining the number of neurons similar to the Sham group. It is not known for sure whether treatment with Butyrate promotes the protection of enteric neurons, preventing them from suffering cell death, or whether it promotes recovery of these neurons through neurogenic stimuli. The possibility of simultaneous protection and recovery cannot be discarded.

Contrary to what happens with neurons, there was an increase in the number of enteric glia in the Colitis group compared to the Sham group. This increase in the number of glia may have occurred through a neuron protection mechanism, where enteric glia would increase their number to compensate for the reduced number of enteric neurons, increasing the glial support offered to the neurons that remained in the nervous plexuses. Furthermore, it has been documented that pro-inflammatory cytokines lead to an increase in GFAP expression in enteric glia and that these cells are increased in tissues with ulcerative colitis [55,56]. The number of glia in the Butyrate group remained similar to the number of glia in the Sham group. Although supposedly, they do not have the GPR41 receptor, treatment with Butyrate may have had indirect actions on the glia through the protection of enteric neurons or even indirectly through the entry of this SCFA into the cytoplasm of these glial cells. It is not known whether SCFA would have any kind of effect on the differentiation of neural precursor cells into enteric glia.

Area analysis of nNOS-ir and ChAT-ir neurons demonstrated cellular alterations that could be occurring in experimental ulcerative colitis. The increase in the cytoplasmic area of nNOS-ir neurons in the Colitis group may be related to mechanisms of cell death, particularly necrotic events [57,58]. However, it is also possible that this increase is due to increased cytoplasmic content, pathological findings typical of cell degeneration, or even an increase in cytoplasmic organelles. On the other hand, the reduction in the total area of ChAT-ir neurons in the Colitis group suggests that, at least in the TNBS protocol and with an interval of 7 days after acid injection, enteric neurons may be affected differently by intestinal inflammation. It is also possible that the mechanisms of cell death/degeneration are different between the types of ENS neurons and that the production of peroxynitrites by nNOS-ir neurons interferes with this observed difference between the areas of nitrergic and cholinergic neurons [54].

In relation to the CTCF analyses, the reduction observed in GPR41-ir and ChAT-ir neurons in the Colitis group is consistent with the loss of these same neurons in this group. Similarly, the increase in CTCF in the GFAP-ir glia in the Colitis group may be related to the increase in the number of glia in this group or to a greater expression of GFAP, which occurs in the face of the inflammatory stimulus [50]. Additionally, the increase in CTCF in GPR41-ir neurons from the Butyrate group may signal a greater activity and/or expression of this receptor compared to treatment with Sodium Butyrate. Regarding nNOS-ir myenteric neurons, the increase in the total area of these neurons may have masked the reduction in CTCF in the colitis group, similar to the other neurons.

The better results observed in this work through treatment with Butyrate may be related to cellular signaling mechanisms triggered after binding these SCFA to GPR41 receptors [16,23,59]. The activation of this receptor promotes the inhibition of the enzyme adenylate cyclase, inhibiting cAMP and protein kinase A [60,61]. These intracellular alterations culminate in the reduction of pro-inflammatory mediators, increase in anti-inflammatory mediators, inhibition of nuclear translocation of NF-κB, and increase in the integrity of the intestinal barrier [16,62,63]. Additionally, mechanisms related to the inhibition of histone deacetylases by Butyrate may be involved in the improvement of intestinal inflammation and the protection of enteric neurons [24,64,65].

The irregular distal colon morphology in the Colitis group analyzed through H&E staining, as well as the reduction of goblet cells, observed using PAS staining, are consistent with histopathological analyzes of ulcerative colitis, frequently found in biopsies of patients with IBD [3,66]. Goblet cell depletion and mucus production reduction seem to be important keys to the pathophysiology of IBD since animals deficient for the Muc-2 gene develop spontaneous colitis [67].

The histological preservation/protection observed in animals treated with butyrate can be explained due to the reduction of pro-inflammatory mediators, but also the particular ability of butyrate to act at the mucosal level, strengthening the integrity of the intestinal barrier [68,69]. The SCFA, mainly Butyrate, manage to increase the integrity of the intestinal barrier through activation of GPR41 receptors or by inhibition of histone deacetylases, promoting an increase in mucous secretion by goblet cells, increase in the expression of intercellular junction proteins; they are also known as tight junctions, such as claudin-1, occludin, Zonula Occludens-1 and Junctions adhesive molecules [24,63,70,71].

## 5. Conclusions

This work demonstrated the colocalization of myenteric neurons nNOS-ir and ChAT-ir the GPR41 receptor and the absence of colocalization of the GPR41 receptor with GFAP-ir enteric glia. Experimental ulcerative colitis affected enteric neurons and glia, and treatment with Butyrate was able to protect myenteric neuron loss and attenuate clinical and histological effects caused by experimental ulcerative colitis. In this sense, Butyrate can be a promising therapeutic tool for the treatment of IBD.

## Figures and Tables

**Figure 1 cells-12-01672-f001:**
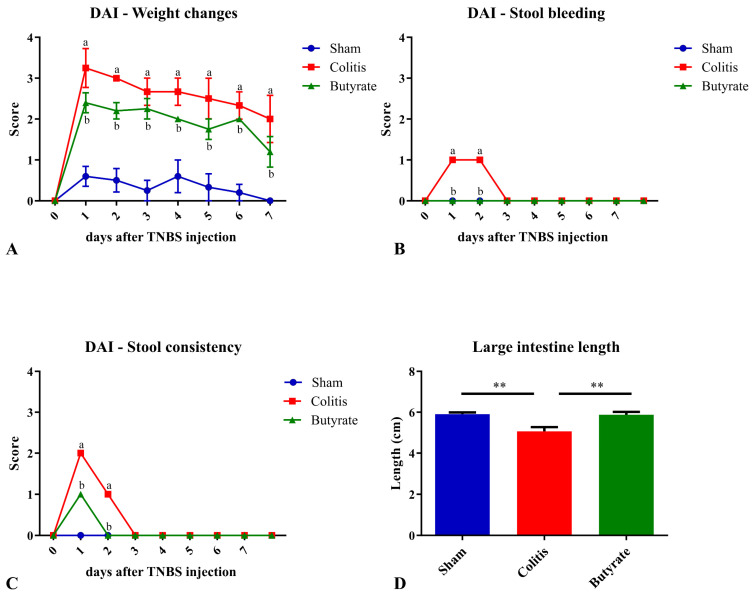
Disease activity index (DAI) scores and large intestine length of the animals in Sham, Colitis, and Butyrate groups. (**A**) Weight changes scores; (**B**) Stool bleeding; (**C**) Stool consistency; (**D**) Large intestine length. Data are from *n* = 5 animals per group and are expressed as mean ± standard error of scores assigned according to Table 1. ^a^ Colitis group compared to Sham group (*p* < 0.05); ^b^ Butyrate group compared to Colitis group (*p* < 0.05). ** *p* < 0.005.

**Figure 2 cells-12-01672-f002:**
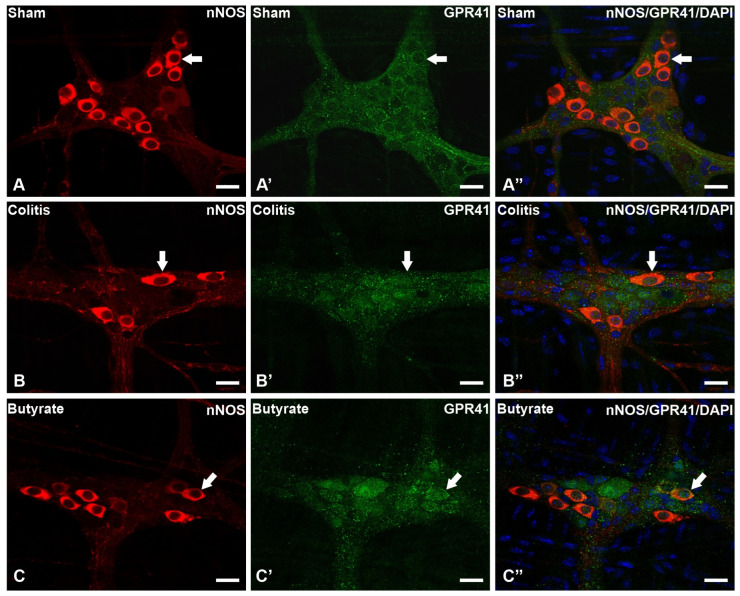
Triple labeling of nNOS-ir neurons (red), GPR41-ir receptor (green), and cell nuclei with DAPI (blue) from the myenteric plexus of the distal colon of mice. (**A**–**A”**) Sham group; (**B**–**B”**) Colitis group; (**C**–**C”**) Butyrate group. Single arrows demonstrate the colocalization of nNOS-ir and GPR41-ir neurons. Bars: 20 µm.

**Figure 3 cells-12-01672-f003:**
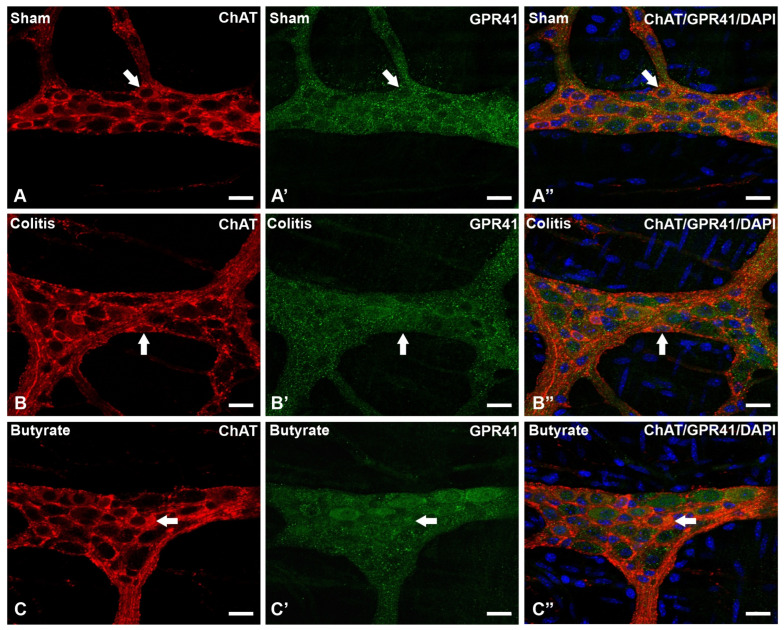
Triple labeling of ChAT-ir neurons (red), GPR41-ir receptor (green), and cell nuclei with DAPI (blue) from the myenteric plexus of the distal colon of mice. (**A**–**A”**) Sham group; (**B**–**B”**) Colitis group; (**C**–**C”**) Butyrate group. Single arrows demonstrate the colocalization of ChAT-ir and GPR41-ir neurons. Bars: 20 µm.

**Figure 4 cells-12-01672-f004:**
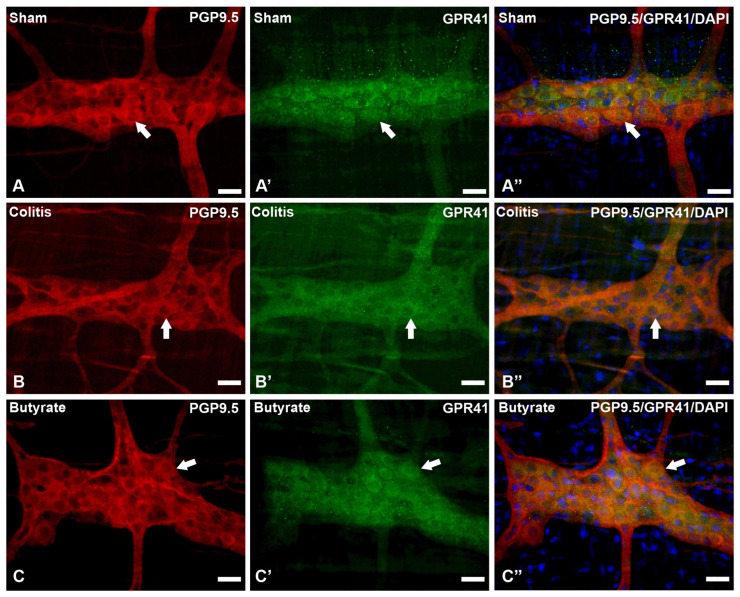
Triple labeling of PGP9.5-ir neurons (red), GPR41-ir receptor (green), and cell nuclei with DAPI (blue) from the myenteric plexus of the distal colon of mice. (**A**–**A”**) Sham group; (**B**–**B”**) Colitis group; (**C**–**C”**) Butyrate group. Single arrows demonstrate the colocalization of PGP9.5-ir and GPR41-ir neurons. Bars: 20 µm.

**Figure 5 cells-12-01672-f005:**
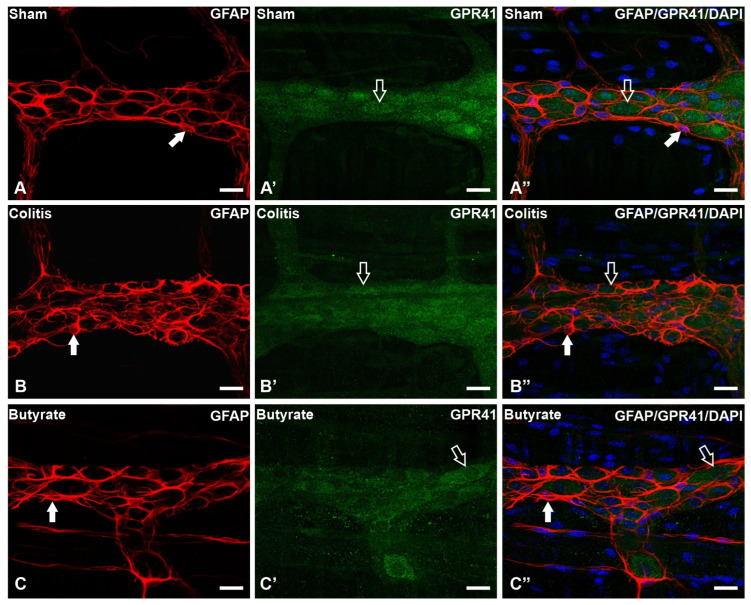
Triple labeling of GFAP-ir enteric glial cells (red), GPR41-ir receptor (green), and cell nuclei with DAPI (blue) from the myenteric plexus of the distal colon of mice. (**A**–**A”**) Sham group; (**B**–**B”**) Colitis group; (**C**–**C”**) Butyrate group. Filled arrows demonstrate GFAP-ir glial labeling. Single arrows demonstrate the labeling of the GPR41-ir receptor. Bars: 20 µm.

**Figure 6 cells-12-01672-f006:**
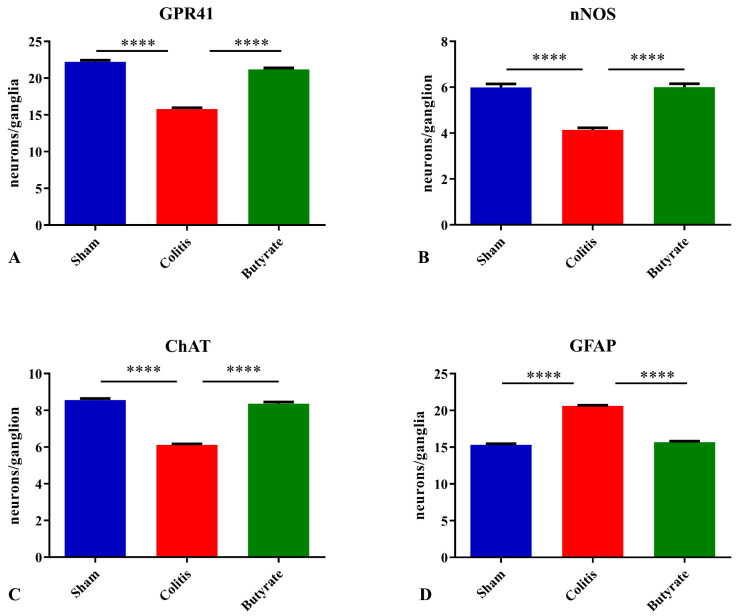
The density of myenteric neurons (neurons/ganglion) immunoreactive to (**A**) GPR41, (**B**) nNOS, (**C**) ChAT, and (**D**) density of enteric glia (glia/ganglion) immunoreactive to GFAP per ganglion in the Sham, Colitis, and Butyrate groups. Data are from *n* = 5 animals per group, obtained by counting 50 ganglia per animal in each group, and are expressed as mean ± standard error of the number of neurons and glia per ganglion. **** *p*< 0.0001.

**Figure 7 cells-12-01672-f007:**
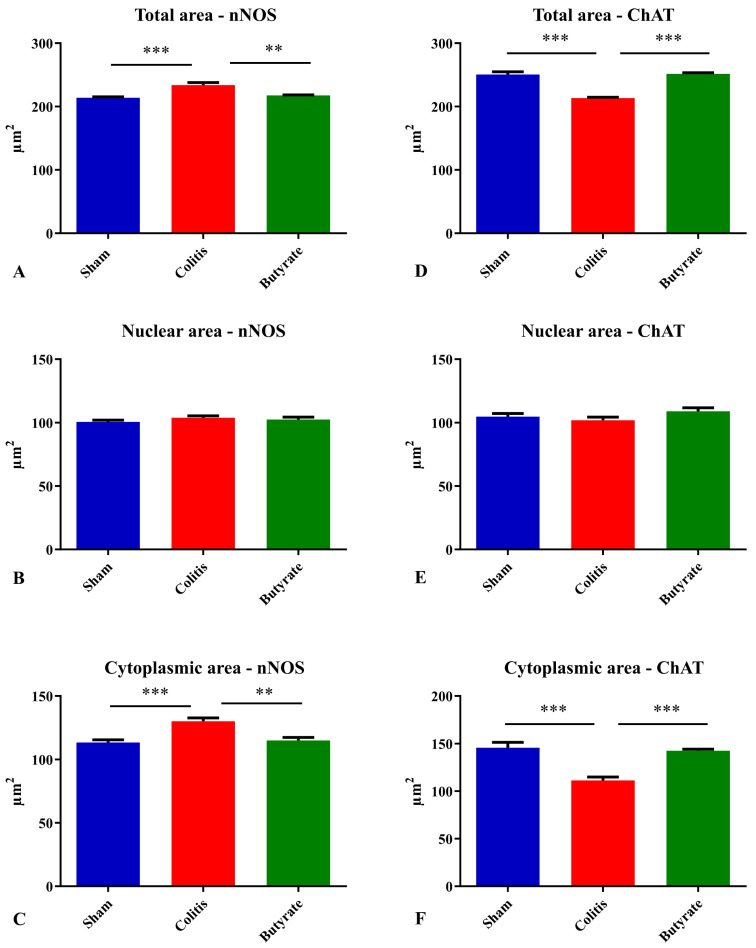
Analysis of the (**A**) total area of nNOS-ir neurons, (**B**) nuclear area of nNOS-ir neurons, (**C**) cytoplasmic area of nNOS-ir neurons, (**D**) total area of ChAT-ir neurons, (**E**) nuclear area of ChAT-ir neurons, (**F**) Cytoplasmic area of ChAT-ir neurons from Sham, Colitis, and Butyrate groups. Data are from *n* = 5 animals per group, obtained by measuring the area of 125 neurons and their respective 125 nuclei per animal, and are expressed as mean ± standard error of area (µm^2^). ** *p* < 0.01; *** *p* < 0.001.

**Figure 8 cells-12-01672-f008:**
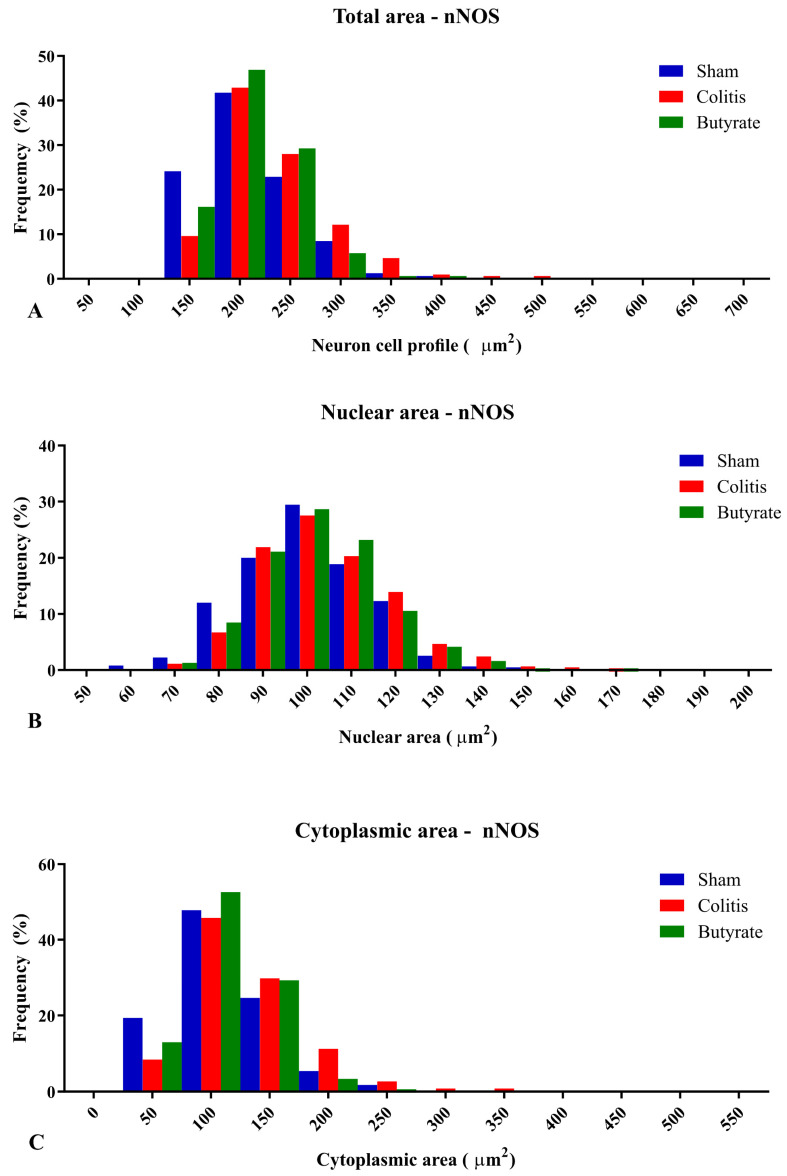
Histogram of the frequency distribution of (**A**) the total area, (**B**) nuclear area, and (**C**) cytoplasmic area of the nNOS-ir neurons of the myenteric plexus of the distal colon of animals from the Sham, Colitis, and Butyrate groups. Data are from *n* = 5 animals per group, obtained from 125 nNOS-ir neurons from each animal, and are expressed in frequency (%) and area (µm^2^).

**Figure 9 cells-12-01672-f009:**
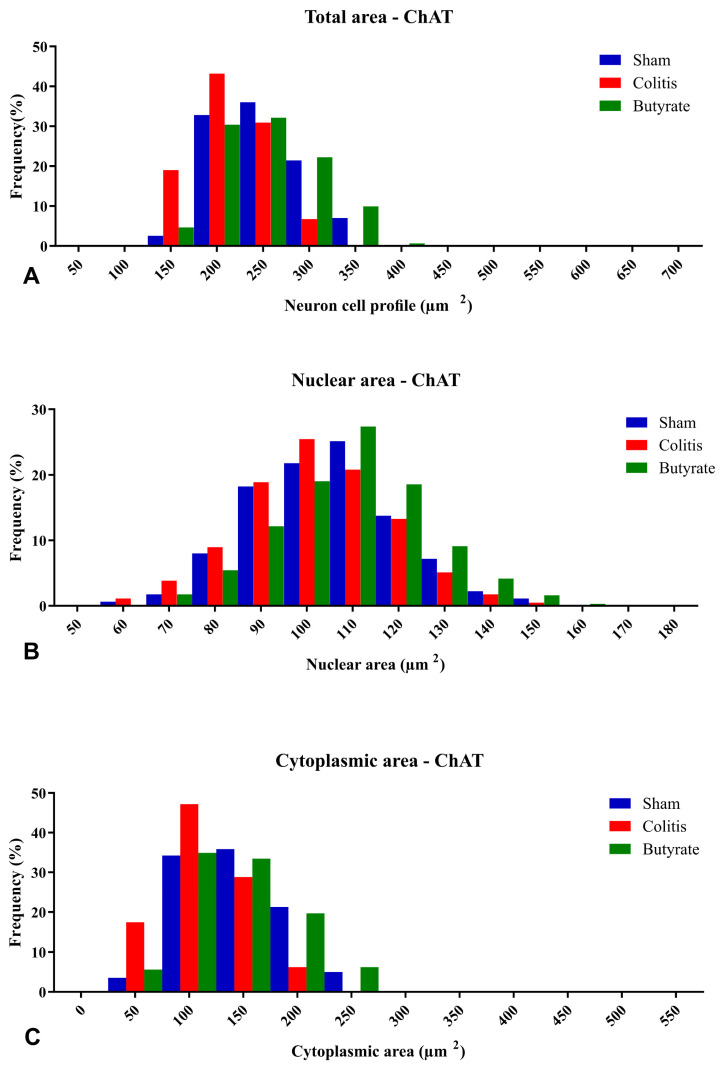
Histogram of the frequency distribution of (**A**) the total area, (**B**) nuclear area, and (**C**) cytoplasmic area of the ChAT-ir neurons of the myenteric plexus of the distal colon of animals from the Sham, Colitis, and Butyrate groups. Data are from *n* = 5 animals per group, obtained from 125 ChAT-ir neurons from each animal, and are expressed in frequency (%) and area (µm^2^).

**Figure 10 cells-12-01672-f010:**
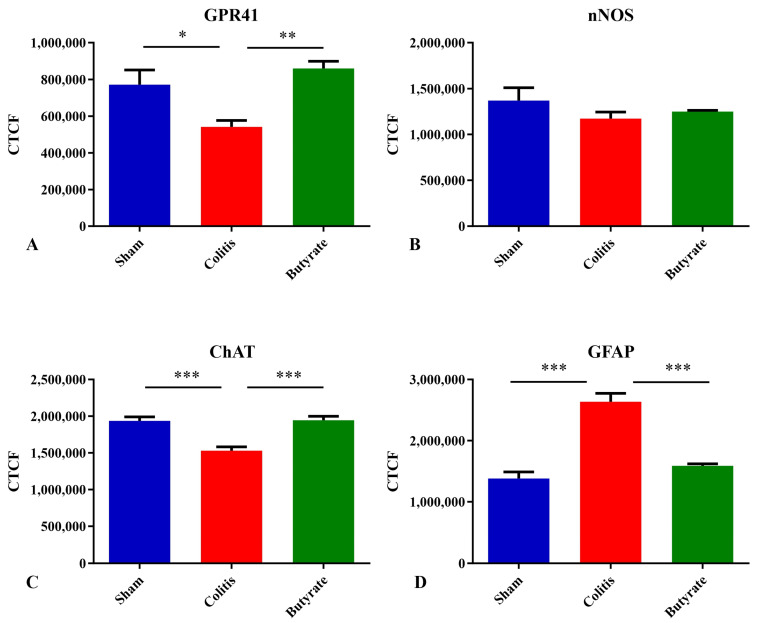
Corrected Total Cell Fluorescence (CTCF) of (**A**) GPR41-ir neurons, (**B**) nNOS-ir neurons, (**C**) ChAT-ir neurons, and (**D**) GFAP-ir glial cells from the myenteric plexus of the distal colon of mice in the Sham, Colitis, and Butyrate groups. * *p* < 0.05, ** *p* < 0.005, *** *p* < 0.001.

**Figure 11 cells-12-01672-f011:**
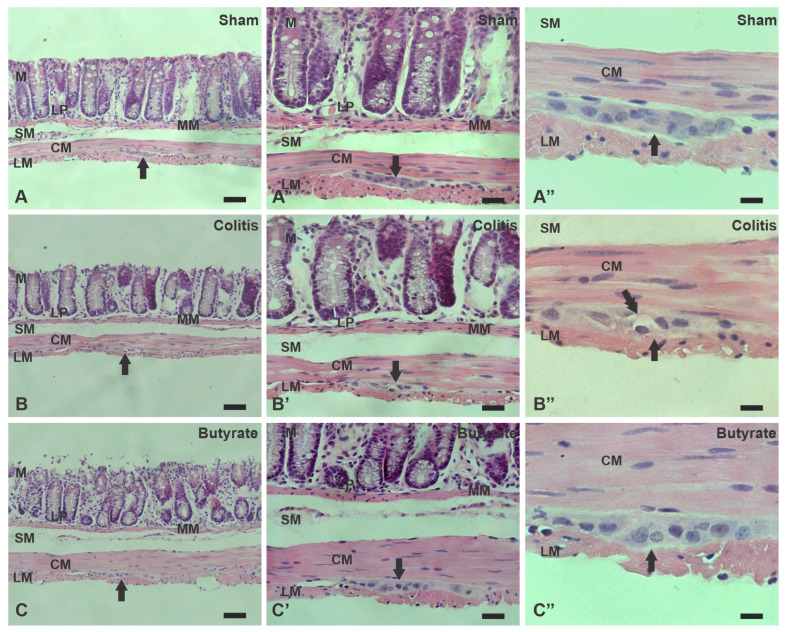
Photomicrographs of sections of the distal colon of mice stained with Hematoxylin and Eosin (H&E) from the Sham (**A**–**A”**), Colitis (**B**–**B”**), and Butyrate (**C**–**C”**) groups. M—Mucosa, MM—Muscularis Mucosa, LP—Lamina Propria, SM—Submucosal, CM—Circular muscle, LM—Longitudinal muscle. Single arrows indicate the myenteric plexus. Double arrows indicate vacuoles in neurons. Bars 50 µm (**A**–**C**); 20 µm (**A’**–**C’**) and 10 µm (**A”**–**C”**).

**Figure 12 cells-12-01672-f012:**
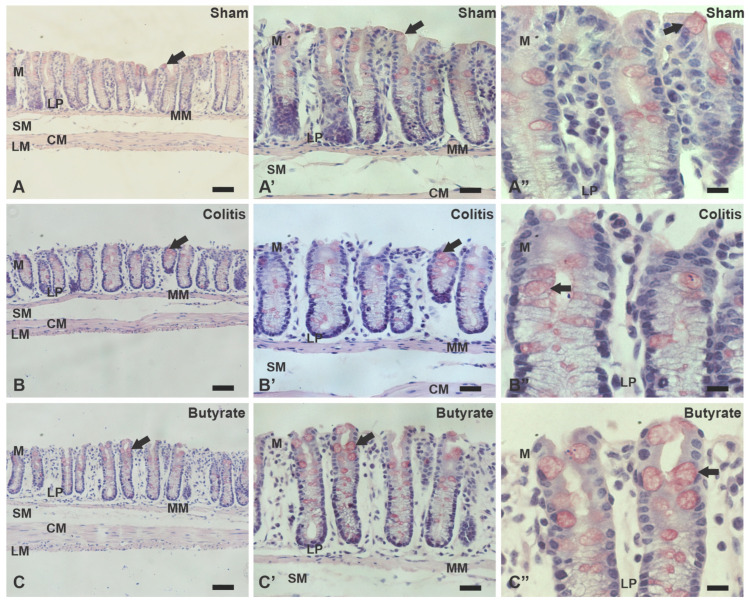
Photomicrographs of sections of the distal colon of mice stained with Periodic Acid Schiff (PAS) of the Sham (**A**–**A”**), Colitis (**B**–**B”**), and Butyrate (**C**–**C”**) groups. M—Mucosa, MM—Muscular mucosa, LP—Lamina propria, SM—Submucosas, CM—Circular muscle, LM—Longitudinal muscle. Single arrows demonstrate goblet cells. Bars 50 µm (**A**–**C**); 20 µm (**A’**–**C’**) and 10 µm (**A”**–**C”**).

**Figure 13 cells-12-01672-f013:**
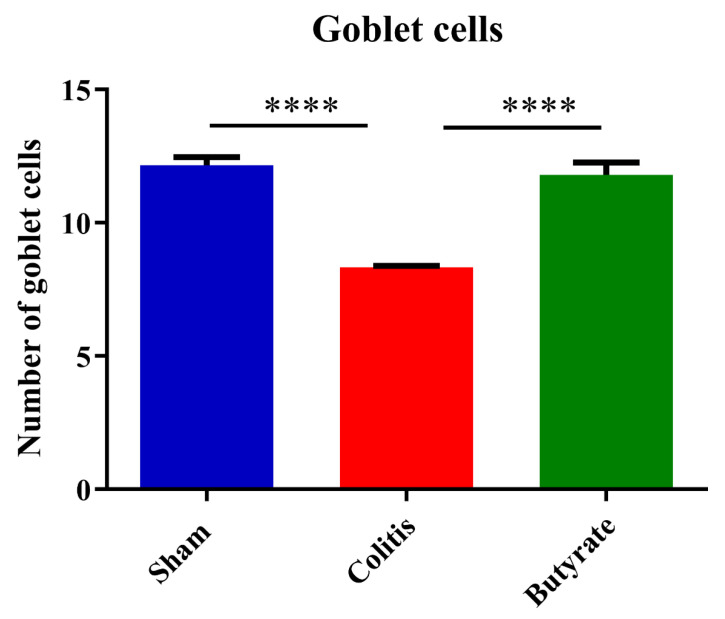
Number of goblet cells in the Sham, Colitis, and Butyrate groups. Data are from *n* = 5 animals per group and are expressed as mean ± standard error of the number of goblet cells in sections of the distal colon of mice stained with Periodic Acid Schiff (PAS). Forty intestinal crypts were counted, in semi-sequential sections, per animal in each of the groups. **** *p* < 0.0001.

**Table 1 cells-12-01672-t001:** Disease activity indices (DAI) are based on percentage weight change, stool consistency, and occult and/or rectal bleeding.

Score	Weight Change	Stool Consistency	Stool and/or Rectal Bleeding
0	<1%	Normal stools	No bleeding
1	1–2%	Soft stools	Mild bleeding
2	2–4%	Soft stools that did not stick to the anus	Moderate bleeding
3	4–6%	Soft stools that stick to the anus	Severe bleeding
4	>6%	Diarrhea	Gross bleeding

**Table 2 cells-12-01672-t002:** Characteristics of the primary and secondary antibodies.

Antigen	Host	Dilution	Source
GPR41	Rabbit	1:200	Sigma
nNOS	Sheep	1:1000	Millipore
ChAT	Goat	1:100	Millipore
GFAP	Goat	1:1000	Sigma
PGP9.5	Guinea pig	1:200	Sigma
Secondary antibodies			
Alexa Fluor 488-conjugated donkey anti-rabbit IgG 488		1:100	Molecular Probes
Alexa Fluor 594-conjugated donkey anti-sheep IgG 594		1:500	Molecular Probes
Alexa Fluor-594 conjugated donkey anti-guinea pig IgG 594		1:100	Molecular Probes

## Data Availability

All data needed are present in the paper.

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
