# Peer review of "Butyrate Protects Myenteric Neurons Loss in Mice Following Experimental Ulcerative Colitis"

_cells, 2023, doi:10.3390/cells12131672_

Round 1

Reviewer 1 Report

In this study, the authors aimed to study the GPR41 in myenteric neurons and analyze the effect of Butyrate in mice submitted to experimental ulcerative colitis. The 2, 4, 6 trinitrobenzene sulfonic acid (TNBS) was injected intrarectally in C57BL/6 mice (Colitis). One group was treated with 100mg/kg of Sodium Butyrate (Butyrate), and the other groups received saline. Animals were euthanized 7 days after colitis induction. Analyzes demonstrated colocalization of GPR41 with neurons immunoreactive (-ir) to nNOS and ChAT-ir, and absence of colocalization of the GPR41 with GFAP-ir glia. Quantitative results demonstrated losses of nNOS-ir, ChAT-ir, and GPR41-ir neurons in Colitis group, and butyrate treatment attenuated neuronal loss. The number of GFAP-ir glia increased in Colitis group, while butyrate reduced the number of these cells. The morphological alterations observed in Colitis group were attenuated in Butyrate group. The treatment with butyrate attenuated the damage caused by experimental ulcerative colitis.

The study is of interest and provides novel findings. It is well designed and clearly presented. However, to further improve the pathogenetic and clinical impact of the study, the authors should suggest and discuss the potential link between their findings and the development of colitis. Previous studies suggested that a dysregulation of the immune system and intestinal microbiota, in subjects with predisposing genetic factors, may be related to the development of IBD. The pathogenic role of immune system is also supported by the development of specific immune response towards some microbial antigens in IBD such as Anti-Saccharomyces cerevisiae antibodies (ASCA). The development of ASCA is considered to reflect the immune response against increased intestinal permeability. Of interest, such a immune response is common in IBD but also in other inflammatory intestinal disorders such as celiac disease as previously demonstrated (Anti-Saccharomyces cerevisiae and perinuclear anti-neutrophil cytoplasmic antibodies in coeliac disease before and after gluten-free diet. Aliment Pharmacol Ther. 2005 Apr 1;21(7):881-7. doi: 10.1111/j.1365-2036.2005.02417.x. ; Anti-saccharomyces cerevisiae antibodies (ASCA) in coeliac disease. Gut. 2006 Feb;55(2):296. PMID: 16407392; ). This is a well-recognized immunological feature characterizing crhonic intestinal disorders such as IBD and celiac disease and is worth mentioning.

 Minor editing of English language required

Author Response

06/08/2023

Dear Reviewer

We would like to thank the editor and the reviewers for their helpful comments and insights regarding our previous submission. We have made the changes discussed below to address the reviewers’ concerns.

            “The study is of interest and provides novel findings. It is well designed and clearly presented. However, to further improve the pathogenetic and clinical impact of the study, the authors should suggest and discuss the potential link between their findings and the development of colitis. Previous studies suggested that a dysregulation of the immune system and intestinal microbiota, in subjects with predisposing genetic factors, may be related to the development of IBD. The pathogenic role of immune system is also supported by the development of specific immune response towards some microbial antigens in IBD such as Anti-Saccharomyces cerevisiae antibodies (ASCA). The development of ASCA is considered to reflect the immune response against increased intestinal permeability. Of interest, such a immune response is common in IBD but also in other inflammatory intestinal disorders such as celiac disease as previously demonstrated (Anti-Saccharomyces cerevisiae and perinuclear anti-neutrophil cytoplasmic antibodies in coeliac disease before and after gluten-free diet. Aliment Pharmacol Ther. 2005 Apr 1;21(7):881-7. doi: 10.1111/j.1365-2036.2005.02417.x. ; Anti-saccharomyces cerevisiae antibodies (ASCA) in coeliac disease. Gut. 2006 Feb;55(2):296. PMID: 16407392; ). This is a well-recognized immunological feature characterizing crhonic intestinal disorders such as IBD and celiac disease and is worth mentioning.”

 R: Thank you very much for your suggestions and consideration. We have added a paragraph with these references to the Discussion. We also did a minor English edit on the manuscript.

Discussion

Line 381-398 "Administration of TNBS diluted in 35% ethanol triggers intestinal inflammation mainly by disrupting the intestinal barrier, as ethanol breaks the epithelial layer and exposes the lamina propria to TNBS itself and intestinal lumen antigens, including microorganisms [40]. Although the pathogenesis of IBD remains uncertain, there are evidences that the development of IBD may be related to dysregulation of the immune system and intestinal microbiota in genetically susceptible patients [41,42]. In individuals with IBD, there is a dysregulation of the intestinal barrier, leading to greater intestinal permeability. This can lead to immune responses against microorganisms of the intestinal microbiota such as the production of anti-Saccharomyces cerevisiae antibodies (ASCA) and the production of perinuclear antineutrophil cytoplasmic autoantibodies (pANCA) [43,44]. ASCA and pANCA have been considered physiological markers for IBDs. In particular, ASCA is more associated with CD and pANCA is more associated with UC [43]. These two markers can be used for the differential diagnosis between CD and UC [45], although they are not exclusive markers of IBD since ASCA is also present in patients with celiac disease [43,44]. In this context, a better understanding of the role of the immune system in response to changes in the intestinal barrier can help to better understand the development of IBD and guide new therapies."

Line  614- 627

  1. Kawada, M.; Arihiro, A.; Mizoguchi, E. Insights from Advances in Research of Chemically Induced Experimental Models of Human Inflammatory Bowel Disease. World J. Gastroenterol. 2007, 13, 5581–5593.
  1. Mentella, M.C.; Scaldaferri, F.; Pizzoferrato, M.; Gasbarrini, A.; Miggiano, G.A.D. Nutrition, IBD and Gut Microbiota: A Review. Nutrients 2020, 12, 1–20.
  2. Leone, V.; Chang, E.B.; Devkota, S. Diet, Microbes, and Host Genetics: The Perfect Storm in Inflammatory Bowel Diseases. J. Gastroenterol. 2013, 48, 315–321.
  3. Granito, A.; Zauli, D.; Muratori, P.; Muratori, L.; Grassi, A.; Bortolotti, R.; Petrolini, N.; Veronesi, L.; Gionchetti, P.; Bianchi, F.B.; et al. Anti-Saccharomyces Cerevisiae and Perinuclear Anti-Neutrophil Cytoplasmic Antibodies in Coeliac Disease before and after Gluten-Free Diet. Aliment. Pharmacol. Ther. 2005, 21, 881–887.
  4. Granito, A.; Muratori, L.; Muratori, P.; Guidi, M.; Lenzi, M.; Bianchi, F.B.; Volta, U. Anti- Saccharomyces Cerevisiae Antibodies ( ASCA ) in Coeliac Disease. Gut 2006, 55, 296–296.
  5. Saibeni, S.; Folli, C.; de Franchis, R.; Borsi, G.; Vecchi, M. Diagnostic Role and Clinical Correlates of Anti-Saccharomyces Cerevisiae Antibodies (ASCA) and Anti-Neutrophil Cytoplasmic Antibodies (p-ANCA) in Italian Patients with Inflammatory Bowel Diseases. Dig. Liver Dis. 2003, 35, 862–868.

Sincerely,

Dr Patricia Castelucci, PhD

Associate Professor

Head of Laboratory of Neurogastroenterology

Department of Anatomy

University of São Paulo

Reviewer 2 Report

A pleasure to read.

An excellent manuscript with the required information necessary to take inflammation beyond the gastrointestinal mucosa.

Author Response

06/08/2023

Dear Reviewer

We would like to thank the editor and the reviewers for their helpful comments regarding our previous submission.

            Comments and Suggestions for Authors

A pleasure to read.

An excellent manuscript with the required information necessary to take inflammation beyond the gastrointestinal mucosa.

 R: Thank you very much for your comments.

Sincerely, 

Dr Patricia Castelucci, PhD

Associate Professor

Head of Laboratory of Neurogastroenterology

Department of Anatomy

University of São Paulo

Reviewer 3 Report

The article was written correctly. The authors demonstrated co-localization for nNOS-ir and ChAT-ir muscle neurons. Experimental induced ulcerative colitis involved intestinal neurons and glia. The use of butyrate in the experimental treatment of IBD has a protective effect on the protection against loss of intestinal muscle neurons and alleviates the clinical symptoms and histological effects caused by experimental ulcerative colitis. The authors suggest that butyrate may be a promising therapeutic tool in the treatment of IBD. The conclusions are in line with the analyses. High levels of inflammation and chronic immune activation affect the gut through a variety of mechanisms, making it more susceptible to inflammation. I have minor remarks:

Missing  of purpose?

Information what kind of animals?

how many weeks old were they when they were induced with UC, CD?

Author Response

06/08/2023

Dear Reviewer

We would like to thank the editor and the reviewers for their helpful comments and insights regarding our previous submission. We have made the changes discussed below to address the reviewers’ concerns.

 Comments and Suggestions for Authors

The article was written correctly. The authors demonstrated co-localization for nNOS-ir and ChAT-ir muscle neurons. Experimental induced ulcerative colitis involved intestinal neurons and glia. The use of butyrate in the experimental treatment of IBD has a protective effect on the protection against loss of intestinal muscle neurons and alleviates the clinical symptoms and histological effects caused by experimental ulcerative colitis. The authors suggest that butyrate may be a promising therapeutic tool in the treatment of IBD. The conclusions are in line with the analyses. High levels of inflammation and chronic immune activation affect the gut through a variety of mechanisms, making it more susceptible to inflammation. I have minor remarks:

Missing  of purpose?

R: Thank you very much for your suggestions and consideration. We have added this information to the manuscript..

Line 76-77, with the purpose of evaluating whether Butyrate has a beneficial effect on ulcerative coli-tis and enteric neurons.”

Information what kind of animals?

R: Thank you very much for your suggestions and consideration. We have added this information to the manuscript.

Line 84, The animals used were  wild-type C57BL/6 mice

how many weeks old were they when they were induced with UC, CD?

R: Thank you very much for your suggestions and consideration. We have added this information to the manuscript.

Line 90-91, 8 weeks-old male wild-type C57BL/6 mice

Sincerely,

Dr Patricia Castelucci, PhD

Associate Professor

Head of Laboratory of Neurogastroenterology

Department of Anatomy

University of São Paulo